# The Classical-Quantum Dichotomy from the Perspective of the Process Algebra

**DOI:** 10.3390/e24020184

**Published:** 2022-01-26

**Authors:** William Sulis

**Affiliations:** Collective Intelligence Laboratory, McMaster University, Hamilton, ON L8S 2T6, Canada; sulisw@mcmaster.ca; Tel.: +1-905-772-7218

**Keywords:** classical-quantum boundary, contextuality, process algebra, generativity, information, locality

## Abstract

The classical-quantum dichotomy is analyzed from the perspective of the Process Algebra approach, which views fundamental phenomena through the lens of complex systems theory and Whitehead’s process theory. Broadly, the dichotomy can be framed in terms of differences in ontology (phenomena and their behavior) and differences in epistemology (theoretical languages used in their description). The Process Algebra posits a reality, generated by processes, whose fundamental characteristics include becoming, generativity, transience, locality, and contextuality. From this perspective, the classical-quantum dichotomy appears to be a false dichotomy—it arises because of stereotyped, strawman-like depictions of what it means to be classical or quantum. A more careful examination reveals that reality is unitary, that whether a system behaves in a quantum or classical manner depends upon its particularities, in particular, whether it is complex or not, and how information flows govern its dynamics.

## 1. Introduction: The Classical-Quantum Boundary Is Sharp?

The classical-quantum dichotomy is often referred to as the classical-quantum boundary, as if there exists a sharp demarcation, on one side of which phenomena are to be described in quantum terms, while on the other side phenomena are to be described classically. Such a distinction hinges, however, upon precisely what is meant by the terms classical and quantum. Broadly speaking, these terms may be understood with respect to two categories: ontological and epistemological.

Here, ontology refers to the phenomenology and behaviors associated with the entities so described by each term. In contrast, epistemology refers to the formal languages and theories used to describe entities along with the collection of devices and procedures used to interrogate them, all inextricably mangled together [1]. That is, a particular entity may be called classical or quantum depending upon whether one or more of its aspects or behaviors fits within the classical or quantum ontology and epistemology. Likewise, it may be called classical or quantum depending upon whether the formal language or theory used to describe it is classical or quantum. Many more categories of distinctions might be postulated, but these two broad categories suffice for the purposes of this article, which is meant to be illustrative, not exhaustive.

In discussing the quantum-classical dichotomy in physics, it is necessary to be precise as to which of these two categories of distinctions one is addressing. Let us first focus on ontology and on epistemological issues related to measurement, since they tend to go hand in hand. In physics, the attribution of the term classical has primarily been to those entities termed objects [2,3]. An object is an entity possessing a specific set of characteristics, modeled, for the most part, upon macroscopic scale, inanimate matter. An object is thought to exist complete in itself, independent of the environment within which it is situated. It possesses properties, whose values may be attributed to the object itself, and which may be determined through a procedure of (non-disturbing) measurement. The measuring apparatus merely ascertains the value of a given property—it plays no role in its determination [4]. These properties are precise and singular, meaning that an object may not simultaneously possess two or more distinct values of the same property. Measurements may be carried out separately or collectively and it is possible to obtain a complete set of measurements covering all properties. These values are generally continuous in range, but not necessarily so. Some properties are fixed or intrinsic to an object (like mass, density, charge) while some are variable or relational (like position, momentum, and energy). The value may depend upon the observer but is fixed for that observer. Objects appear to separate ontology and epistemology.

An object is defined or identified by its intrinsic properties, whereas its states are given by its variable properties. An object cannot manifest two distinct states simultaneously. An object is passive in relation to its environment. It reacts whenever the environment acts upon it, but it does not initiate action upon the environment. Interactions with the environment are spatially and temporally localized, and generally continuous in their altering of properties. Interactions alter variable properties of objects, but not their intrinsic properties or structure—that would result in new objects. An object which is spatially localized (idealized as a point) is termed a particle. An object which is spatially distributed may be termed a body, medium, or field depending upon its structure. A body and a medium are thought to be composed of particles whereas a field is not. It is a separate category of object. Extended objects may interact with their environment at multiple sites simultaneously. Examples of particles include atoms in a gas or billiard balls interacting ballistically; examples of bodies include rocks, gyroscopes, gels, putty; examples of waves include water and sound waves; examples of fields include electric, magnetic, and gravitational fields.

Motion refers to changes in state over time. Motion may be localized (particle-like) or distributed (wave-like). Notice that states may sum, for example, position, momentum, amplitude, but measurements of the object only reveal the value of the sum, never its individual contributions, so while states may sum, an object can never be measured in two distinct states simultaneously. Wave states may interfere, while particle states do not.

These properties of object give rise to the notion of objectivity, a separation between an observer and that which is being observed. Subjectivity, on the other hand, applies in situations when such separation is not possible (here I am referring solely to the act of observation, not to psychological subjectivity).

Quantum entities are quite different. They appear to blur the distinctions between ontology and epistemology. Quantum entities cannot be easily defined independent of their environment. Quantum entities possess the same range of properties as do objects, but these properties appear to be inextricably linked to the process of measurement. Quantum measurements often result in a discrete range of values for a property. A quantum entity may manifest two or more distinct states simultaneously and yield distinct values for the same measurement. Measurements may be carried out individually but may not always be carried out simultaneously. Associated with measurement is the concept of (non)-commutativity. Non-commutative measurements may not be measured simultaneously, and even consecutive measurements may destroy the results of previous measurements. It is not possible to obtain a complete set of measurements. A quantum entity is not localized (at least is not point-like). A quantum entity may sometimes behave like a particle while at other times like a wave (wave-particle duality). Interactions are generally discrete in their altering of properties. Even spatially extended entities like electromagnetic fields interact with their environment discretely and locally. In the sense above, quantum entities are subjective. Examples of quantum entities are fundamental particles, atoms (when subject to interaction with light or at extremes of temperature and pressure).

Let us now turn to epistemological issues related to the formal languages and mathematical structures used to describe entities. Classical entities are modeled using finite dimensional vectors (particles), geometric solids (bodies), or continuous functions (waves, fields) over a real space. Each entity is given a specific collection of properties, each having a single value. Their variable properties are modeled by scalars (energy) or vectors (position, momentum). The values of these properties are explicit in the description of the entities. States may sum, but only the sum is observable. Motion is described most commonly using differential equations involving these vectors or functions. These fundamental equations are deterministic. Statistical methods are often used when dealing with large numbers of entities, but their individual motion is still considered to be deterministic—statistics are used to compensate for a lack of detailed knowledge.

Quantum entities, on the other hand, are modeled using infinite dimensional vectors and operators in a Hilbert space. These vectors give rise to probabilities via the Born rule and are fundamental to the description of quantum entities. The equations of motion involve relationships among operators. Intrinsic properties appear as parameters. Variable properties are in a sense implicit, appearing as a result of the action of an operator on one of these infinite dimensional vectors, a somewhat ad hoc procedure termed measurement. States superpose, yet when observed, the properties of the individual states are preserved, and so measurement of a given property results in different values which are statistically distributed in a predetermined manner. The act of measurement can change the state of an entity. These infinite vectors exhibit interference in certain cases, giving rise at times to particle-like, and at other times to wave-like effects. Another feature of quantum entities is that measurements do not commute, which leads to contextuality.

A deeper difference appears when one closely examines the structure of the probability theory associated with each domain. In the classical domain, the probability theory is that of Kolmogorov [5]. Consider a case in which one has a system upon which one may perform two different measurements *a*, *b* resulting in the dichotomous outcomes *a*={*a*_1_,*a*_2_} and *b*={*b*_1_,*b*_2_}. If these measurements are performed on a classical entity, Kolmogorov theory shows that the sum of probabilities takes the form $P\left(b=\beta \right)\;=\;\sum_{a_{i}}\;P\left(a=a_{i}\right)P\left(b=\beta |a=a_{i}\right)$.

On the other hand, if these measurements are carried out on a quantum system, then probabilities are calculated by means of Born’s rule. The sum of probabilities then takes the form $P\left(b=\beta \right)\;=\;\sum_{a_{i}}\;P\left(a=a_{i}\right)P\left(b=\beta |a=a_{i}\right)+2\cos\Theta \times \sqrt{P\left(a=a_{1}\right)P\left(b=\beta |a=a_{1}\right)P\left(a=a_{2}\right)P\left(b=\beta |a=a_{2}\right)}$. where Θ is a phase factor representing the degree of interference between alternatives (context effect) [6,7,8].

Contextuality in quantum systems is not merely the fact that the act of measuring can affect the presence and values of any property being measured (an expression of non-commutativity of measurement operators), but also the fact that certain correlations among measurements may violate one or more inequalities (the Bell inequalities and variants), which is not possible given a Kolmogorov probability. There is an entire field of research devoted to the study of the inequalities and their bounds. In the case of one such inequality, the CHSH inequality, the bound for classical systems is said to be 2 (due to Bell [9]), for quantum systems it is $2\sqrt{2}$ (due to Tsirelson [10]) while in general it is 4. This brief review suggests that the major differences distinguishing between classical and quantum entities are: superpositions of states, non-Kolmogorov probability, non-commutativity of measurements, contextuality, and discreteness of interactions. These differences in probability structure and the presence or absence of contextuality are sometimes viewed as defining the differences between classical and quantum [11,12,13].

## 2. The Classical-Quantum Boundary Is Fuzzy

As portrayed in much of the physics literature on this matter, the classical-quantum boundary is hard, impermeable, and unassailable [11,12,13]. The classical world is governed by Kolmogorov probability—period. The quantum world is governed by the Born rule, a non-Kolmogorov probability—period. If the classical realm phenomenologically consisted solely of classical objects, each existing independent of one another and interacting solely classically, then this assertion equating classicality and Kolmogorov probability would appear to be correct. After all, these assumptions appear implicitly or explicitly in most expositions of Kolmogorov probability theory [5].

Classical entities need not be unitary wholes—more often than not, they are composed of parts, lesser wholes which interact with one another to form the larger whole which is the entity. Furthermore, composite entities, when suitably formed, may cease to be simple (for example described merely by mass, center of mass, moment of inertia) and instead become complex [14,15,16]. Unfortunately, there is no simple definition of complex. To some degree it lies in the eye of the beholder and the particular focus of interest—it can be linked to composition, structure, dynamics, description, the capacity for agency, or adaptation [14,15,16,17,18]. Most significant for the argument below is that complex systems may become capable of emergence. One concept of emergence is that it refers to a situation which admits description by multiple, mutually irreducible semantic frames [17]. A similar idea was expressed by Rosen, who said that a system was complex if it could not be understood through a Newtonian formalism [18]. More will be said about semantic frames later, but the depiction of classicality in physics is more of a cartoon than a veridical portrayal.

When one looks out into the classical realm, one sees much more than inanimate matter. There is a vast, messy, complex world of organisms, which cannot and should not be ignored. Yet physics has done that throughout most of its history [19]. However, once one looks closely at the nature and behavior of organisms, the boundary between the classical and the quantum realms appears a lot fuzzier and more porous.

Organisms are profoundly different from classical objects. They are transient entities—becoming, being, fading away are fundamental to their character. They are born, they live, and they die. Throughout their lifespan, they actively interact with their environment. They act upon it, shaping it in ways which elicit, support, and sustain functionality [20,21]. This is especially true of neural and collective intelligence systems such as social insect colonies [22,23,24,25]. Behavior is generated, not merely elicited. The agents that participate in the generation of some behavior may be different with each instantiation of the behavior, as may be their pattern of dynamical interaction [26]. This has been observed among place cells in the rat hippocampus [27] and in human long-term memory [28]. Biological systems are always open systems—they are constantly exchanging components and interacting with their environment. Unlike classical objects, whose structure is assumed to be fixed, the structure and composition of organisms is constantly changing. They begin life with a phase space defined by a single cell, grow to a phase space composed of trillions of cells, many of which do not even share DNA with the central organism, and end up as a dead husk of inert cells.

Like quantum entities, it is often difficult to know where the boundaries for organisms lie. Organisms frequently rely upon commensal relationships with other organisms with whom they do not share DNA (previously thought to be a defining characteristic of an organism). Neural function depends upon and is influenced by the actions of intestinal microbiota and neuroimmunohumoral peptides released by somatic organs. Psychological functions rely heavily upon structure and regularities in the environment. This is particularly true of collective intelligence systems but is true of neural systems as well [22]. The dynamics of neural and collective intelligence systems involve both local (particle-like) aspects (neurons and worker ants) and non-local (field/wave-like) aspects (neuropeptides, neurohumoral factors, volume-distributed neurotransmitters, pheromones, stigmergic effects) [22]. There is not one or the other but rather a complex admixture of both.

Organisms are in continual interaction with their environment through their behavior. They react to their environment, but they also initiate actions upon and interact with their environment. As a result of such interactions, their internal structure and dynamics undergo change. This results in a form of contextuality, which Dzhafarov has termed “Contextuality by default” [29]. Probability distributions associated with behavior, particularly those distributions related to the underlying neural and somatic systems responsible for the generation of behavior, are contextual, meaning that such distributions must be linked to the particular context in which they were generated. One consequence of this is that results obtained in one context need not generalize to other contexts. Biomedicine has learned the potentially catastrophic consequences of ignoring this particular form of contextuality [30]. As organisms adapt over time to experience, their internal dynamics and structure change with the result that the probability distributions describing their behavior also change, that is, their probabilities are non-stationary. This non-stationarity may occur over long time scales, but it may also occur on very short time scales [6,20,21,22]. It has long been known in psychology and sociology that the order of presentation of questions matters [31]. This is a form of non-commutativity of observations. A form of contextuality analogous to that demonstrated by inequality violations has been shown to exist macroscopically [32], in psychological experiments at both the group [33] and individual level [34] and may appear in some collective intelligence systems [22].

Unless one wishes to adopt a set of conceptual blinders, the behavior of organisms should convince one that the boundary between the classical and the quantum is not determined by the presence or absence of contextual effects, whether contextuality by default or true contextuality. Equally, it cannot be determined by the presence or absence of non-commutativity of observations. These may be thought to distinguish between the world of classical objects and that of quantum entities; however, a simple example suggests that even that need not be true. Consider the following (taken from [7]).

There is a 2 × 2 LEGO mounting block fixed inside a sealed box. Within the box is a bag containing a 1 × 1 LEGO block and a 2 × 2 LEGO block. There is a dial on the outside of the box which reads 0, 1, 2. When the dial is set, a reading is taken of the plate and a light turns on, corresponding to whether there is no block on the plate (0), a 1 × 1 block (1), or a 2 × 2 block (2). The examiner cannot look in the box and in fact has no knowledge of the contents of the box. They can only switch the dial and note whether or not a light appears. In another room, a researcher can remotely arrange whatever they like on the plate: no block, a 1 × 1 block, or a 2 × 2 block and they change the arrangement immediately following each observation by the examiner. Clearly, the probabilities of no block, a 1 × 1 block, or a 2 × 2 block are all 1/3. Therefore, for the examiner, the probabilities of obtaining a light for 0, 1, 2 are all 1/3.

Now let us change the game slightly. The researcher is now permitted to take no action, place a 1 × 1 or a 2 × 2 block on the plate, or to couple the 1 × 1 block to the top of the 2 × 2 block and affix this to the plate. Setting the dial to 1 or 2 results in a light so long as the corresponding block is present regardless of whether it is alone or in combination. Note that it is impossible in this arrangement to measure for 1 and 2 simultaneously. Now, what is the probability of there being a light on 1? This probability is 1/2 because there is a 1/4 probability of there being a single 1 × 1 block and a 1/4 probability of there being a 1 × 1–2 × 2 combination. The same holds for the probability of a light on 2, while the probability of a light on 0 remains 1/4. Note now that *P*(0) + *P*(1) + *P*(2) = ¼ + ½ + ½ = 5/4. As far as the examiner is concerned, the outcomes are disjoint, but the sum is not 1.

One might argue that the problem is that the probability distribution has been incorrectly specified. That is, however, a mathematical cheat, because the examiner constructs their probability distribution based upon empirical observation, just as any physicist does. They cannot look inside the box any more than a physicist can. From the point of view of the examiner the space of alternatives was correctly constructed, and they are disjoint. However they must also accept the necessity to introduce an interaction term, or to accept a non-standard form for the calculation of the total probability, namely *P*(total) = *P*(0) + *P*(1) + *P*(2) + I(0,1,2) = ¼ + ½ + ½ − 1/4.

The researcher here plays the role of the external world and there is no communication between researcher and examiner, just as there is no communication between Nature and physicist. The fact that the probabilities do not follow Kolmogorov laws reflects the presence of interaction between components in the second experiment. In other words, they are not behaving like classical objects, even though they are classical objects. That is because interactions which change the nature of entities are not allowed in the classical model, whereas they are ubiquitous among organisms and quantum entities. The difference in probabilities should be a clue that interactions are taking place.

Note that in neither the psychological case, nor in the example of the LEGO blocks, does non-locality play any role in giving rise to the non-Kolmogorov probability or the contextuality. Kolmogorov made the point explicitly in his writings on probability that it was essential to take context into account when assigning a probability distribution to a set of events [5]. The ability to form a joint distribution from a set of individual distributions is principled—it can only take place and be meaningful when specific conditions are met [7,8]. Much existential grief might have been avoided had only more researchers remembered his words.

Five possible sources of distinction between classical and quantum were described previously. These were: superpositions of states, non-Kolmogorov probability, non-commutativity of measurements, contextuality, and discreteness of interactions. The above discussion, although brief, suggests that the presence of non-Kolmogorov probability, non-commutativity of measurements, and contextuality cannot form the basis for demarcating the boundary between classical and quantum. Under certain situations, organisms, and sometimes even classical objects, may show these characteristics. They can only serve to distinguish the two realms if one takes an excessive, arbitrary, and unreasonably restrictive definition of what it means to be classical. However, that would appear to beg the question in that case.

Two characteristics remain: superpositions of states, and discreteness of interactions. Superpositions of states are a feature of the classical world just as they are of the quantum world. The equations of motion describing each realm are generally (with some exceptions) linear, which means that sums of solutions are also solutions. In the classical world, however, when one performs a measurement to determine the state of a system one only ever obtains a single value, which corresponds to the sum over any presumed underlying component states. The situation is different in quantum mechanics, where measurement of a superposition yields (at least when an ensemble of identical systems is considered) a set of values, each corresponding to that of a component state. These quantum superpositions have been observed in single particles and atoms but also in larger structures such as molecules [35,36]. To date, such superpositions have not been convincingly demonstrated in macroscopic bodies or in organisms. However, there is a phenomenon of mixed emotion in psychology, in which two distinct emotions appear to be manifested in an individual as a superposition and probing of that individual yields one or the other emotion over time [37]. This is similar to what is observed when probing superpositions of single photons [38]. While the mechanisms involved are clearly different (and a mathematical treatment of the emotion situation is lacking for comparison), nevertheless there is face validity at a phenomenological level.

This leaves the discreteness of interactions as the only remaining candidate for demarcating the boundary between classical and quantum. It certainly appears as though the effects of interactions appear to range continuously in the classical realm. Nevertheless, again considering neural and collective intelligence systems, there is an abundance of situations in which interactions are governed by thresholds—that is, behavioral responses are elicited in an all or nothing manner [22,23,24,25]. The appearance of gradation in the response, such as in the case of recruitment by workers in a social insect colony, appears only at the colony level and is reflective of the large number of workers that may respond to the trigger. The response of any individual worker to the trigger is all or nothing. Organisms use a host of signals to communicate and to influence behavior. Humans developed language, especially written language, which is by its very nature discrete. Of course, where the quantum realm really stands out is in the interaction of fields with matter, such as light impinging on a photoelectric cell. Classically, the effect of the field should be expressed wherever it manifests (just as a water wave may simultaneously affect anything along its path) but in fact the effect is localized and discrete. It is difficult, even among organisms, to conceive of a situation analogous to this quantum phenomenon.

Physicists are fond of referring to inconvenient conflicts between perceived reality and the predictions of their theories as “illusions”. It is quite possible that the experience of continuity itself is one such illusion. It may be a powerful and useful illusion, it may provide a powerful set of mathematical tools (and results), but as a feature of the natural world, it may nevertheless be illusory. Certainly, whenever one has peered deeply into the structure of any material entity one has only found discreteness. Every entity proves to be composed of bits, which are themselves composed of bits, all the way down to the fundamental level. Only space and time appear to be continuous, yet there is increasing evidence that even that assumption may not be true [39]. Thus, it may turn out that even in the classical world there is a fundamental discreteness—the reason why we perceive it to be continuous is because our human perceptual apparatus is simply incapable of detecting the difference between units.

The above discussion should suggest that perhaps the classical and quantum realms have more in common with one another than one might think at first glance. It is not unreasonable to suggest that if the founders of quantum mechanics had had more experience with the structure and behavior of organisms, or of psychology per se, then many of the arguments about the nature of reality might have been avoided, and the taint of quantum mysticism would not have appeared. The question is not where the boundary lies between the classical and the quantum—that boundary is likely illusory. Instead, the question to ask is under what circumstances a system exhibits quantum-like characteristics and when it exhibits classical-like characteristics instead—are these characteristics scale-based, structural, dynamic, an effect of isolation or openness? Within quantum mechanics the main approaches to these questions have been through the addition of processes such as continuous spontaneous localization [40], decoherence through interaction with the environment [41], or through consistent histories [42]. We turn next to a discussion of the classical-quantum dichotomy through the lens of the Process Algebra.

## 3. Eye of the Beholder

From the perspective of the Process Algebra, there is no singular, separate quantum reality, nor is there a singular, separate classical reality. There is a single, consistent, unitary reality in which different kinds of entities exist. Some of those entities have classical characteristics while some have quantum (and possibly there exist some having different characteristics that have not been encountered as of yet). The common thread linking these entities is the idea of emergence, which comes out of the theory of complex systems, especially complex adaptive systems [14,15,16,17,18,19,20,21,22,23,24,25]. The appearance of delimited tracks in a bubble chamber is an example of a quantum system exhibiting emergent classical-like behavior [43]. One approach to the problem of emergence is the semantic frame [17,44]. The concept of a semantic frame is a generalization of the physicist’s concept of a frame of reference—it provides a system, usually an organism, with the ability to structure its perception of its environment into units that are salient for its survival, and to attach meaning to those units and their actions, which in turn guides its selection of future behavior, enabling it to provide for its needs and fulfill its goals. The semantic frame answers the six basic questions: who, where, what, when, why, and how. The answers to these questions, regardless of their veracity, nevertheless help to determine the future behavior of the system.

Archetypal dynamics, the formal theory of semantic frames [44] posits that complex systems are distinguished by virtue of admitting, at least in part, descriptions of phenomena using multiple, mutually irreducible, semantic frames.

The semantic frame is a concept based upon meaning-laden information. Formal theories, systems of cultures and values, philosophical systems, theologies, all serve as examples of semantic frames. Behavioral transients may serve as the basis for information flow within systems, with salience being expressed through the appearance, as a result of the system’s dynamics, of stimulus-response-type correlations between transients in the environment and transients generated by the system [45]. These transients give rise to a flow of information within a complex system [46]. It has been suggested that this primitive stimulus-response correlation could provide the basis for an intrinsic semantic frame [17,44,45,46].

Neither classical nor quantum mechanics are capable of describing the behavior of all systems in this extended reality. As a trivial example, try poking a dog with your finger and see how well they predict what will happen. A more serious example was provided by Rosen decades ago in a paper in which he discussed a quantum-mechanics-based model of genetic information transfer [19,47]. He showed that the process of genetic information transfer did not possess a Hamiltonian, and thus fundamentally did not admit a description within the framework of quantum mechanics. Energy, entropy, or the meaningless information of physics are limited in the degree to which they determine the behavior of complex systems [46,47].

Another result of Rosen [48] may further illustrate how a semantic frame, by specifying the manner in which an observer interacts with the system, shapes the “reality” presented by that system. Rosen introduced, within a wholly classical setting, the concept of the analogous system, a system whose dynamics stands in analogy with that of another [48]. He showed that there exist universal dynamical systems, in the sense that given any dynamical behavior, it is possible to select a set of observables such that the universal system, interrogated via these observables, produces behavior analogous to the chosen dynamics. The universal system is real in the sense that it has definite states and its own self-consistent, independent dynamics. Nevertheless, an observer, by selecting the manner in which the system is interrogated, in other words, by specifying a semantic frame, will observe any dynamic whatsoever. The observer does not create reality, rather they select out one particular expression of reality through their interaction with the system. Could it be that all of our physical theories bear this relationship to the larger sphere of phenomena which we term “reality”?

The three pillars of modern physics, classical mechanics, quantum mechanics, and general relativity, may all constitute semantic frames. Their effectiveness as theories does not simply depend upon the scale of phenomena to which they are applied, it also depends upon the mathematical structures which are used to express the theory and upon the manner in which an understanding of those structures shapes and guides our relationship to the phenomena being described. Years ago, Mermin warned of the dangers of reification [49], that is, confusing the thing described with its description. This leads to the tendency to think of natural kinds as mathematical objects, instead of keeping in mind that the mathematical object only describes some aspect of the entity. Mathematical objects may derive their properties from the study of natural kinds, but natural kinds do not derive their properties from mathematics but rather from their natures, and their dynamics, and their interactions with the natural kinds around them. The presence of emergence throws a monkey wrench into any attempt to develop a “theory of everything”.

It may have been a mistake for the founders of quantum mechanics to have placed so much emphasis on the Hilbert space formulation. The Schrodinger equation is linear, and therefore the sum of two solutions of the equation is also a solution, but each solution represents a distinct ontological state. Summing two distinct ontological states is a bit like trying to merge an apple with a banana. It might work in special cases, but there is no guarantee that it should work in every case.

As is well known, each pillar is framed using a different form of mathematics: classical dynamics is a dynamic on manifolds, quantum mechanics is a dynamics of probability distributions (and operators) on manifolds, while general relativity is a dynamic of manifolds. Formally, these theories are not compatible. Indeed, they may be irreducible. That is not surprising really if reality is viewed through the lens of complex system theory, and the various entities that form reality are considered to be emergent. In that case, the idea of a classical-quantum dichotomy says more about the nature of the theories used to describe aspects of reality than it does about the reality itself. Indeed, the discussion of the previous section shows that there is a great deal of similarity between the phenomenology of organisms and that of quantum systems. In fact, there is an area of research devoted to exploring how well quantum formalism (the mathematics sans the physical interpretation) may be applied to framing theories about psychological and social systems [50,51]. This, in turn, suggests that the debate about the classical-quantum dichotomy (and especially the demarcation of any boundary) is more about determining the limits of the explanatory scope of competing mathematical theories than it is about the reality that those theories purport to describe. Reification may account in part for this confusion.

The Game of Life provides a nice, clear example of emergence, at least in a formal setting. Conway’s Game of Life [52] provides a simple illustration of the problem. The Game of Life is a two-dimensional, two state, eight neighbor, cellular automaton with simple rules. If the state of a cell is 1 (alive), then its next state is 1 if it has exactly 2 neighbors having state 1, otherwise it is 0 (dead). If the state of a cell is 0, then its next state is 1 if and only if it has exactly 2 or 3 neighbors, otherwise it is 0. These rules are so simple that a child can follow them.

Conway proceeded to demonstrate something quite remarkable and profound [52]. He showed that certain configurations of states (independent of the particular cells manifesting them) could be organized on the automaton space in such a way that as the underlying automaton dynamics unfolded, these configurations would undergo a consistent succession of changes. These dynamical patterns could be meaningfully and consistently interpreted in terms of packets of information and logical operations on those packets. He was able to show that these configurations could be used to simulate the actions of a Universal Turing machine. Note that the individual cells, themselves, did not function as Universal Turing machines. Their actions were deterministic, and their dynamics fixed. Nevertheless, configurations of states (initial states) could be formed on the automaton space corresponding to the structure of this Universal Turing machine and some initial data, and then allowed to evolve under the automaton dynamics until a particular configuration appeared (a halting or final state), at which point the global configuration could be examined and the result of the computation determined. The dynamics of these configurations could be understood entirely within the theory of Universal Turing machines, without any reference to the underlying automaton dynamics. Moreover, there is an infinity of configurations whose evolutions do not follow the theory of Universal Turing machine. The result is not general; it is particular.

Note too that the Universal Turing machine perspective does not apply to configurations of cells—it applies to configurations of states. The particular cells which support these configurations change over time. The cells become elements of an abstract “space-time” while the configurations of states become the “entities” which exist within that space-time. These entities thus can move around in the space-time and interact within the space-time.

That the Game of Life supports a Universal Turing machine is not directly provable from the cellular automaton rules. It is an emergent property satisfied by particular configurations of cellular automaton states. Neither the global rules nor the rules are reducible to one another. Thus, the cellular automaton and the Universal Turing machine perspectives constitute separate, mutually irreducible semantic frames—they describe the relevant entities, where they are, when they are situated, how they behave, and further constrain the manner in which an observer must interact with the system in order to preserve the consistency of the observed phenomena with the frame. It is worth pointing out that the state patterns which comprise some particular computation must be imposed on the Game of Life by an external observer, one who acts in a manner consistent with the Universal Turing machine semantic frame. For example, one cannot simply impose any arbitrary state pattern on the cells of the automaton and expect the subsequent evolution to remain consistent with the interpretation provided by the Universal Turing machine frame. The cells onto which such patterns are imposed are fungible—their role is not a feature of the automaton itself, similar to what is observed in other living systems such as neural systems and collective intelligence systems as noted previously.

In physics, it is often the case that experiments are constructed, guided by the requirements of some theory, and particular sets of measurements are carried out, the results later interpreted within the theory. Indeed, arguments within quantum information theory are often crafted using computational semantic frames. Quantum mechanics itself was originally formulated as a theory for determining the outcomes of measurements, not as a theory of the phenomena which give rise to measurements [53]. In a strict sense, quantum mechanics does not tell us how natural phenomena behave in interaction with one another “in the wild”, but only what kinds of measurements a system will yield when interrogated within the lab. In many respects, classical theory is similar, but the ability within the classical framework to attribute specific properties directly to the systems being modeled enables one to say that the theory is about the system itself, not merely our interrogations of the system.

## 4. Classical and Quantum Are Bound(ary) Less?

It would appear that both classical and quantum mechanics (and perhaps general relativity as well) relate to the real world analogous to the relationship between the Universal Turing machine frame and the cellular automaton frame of Life. That is to say, each constitutes a semantic frame. As we have seen, the manner in which one interrogates a system may have a profound influence upon the behavior that one receives in return. Rather than saying that a phenomenon is classical, or *is* quantum, (or *is* neither), it would be better to assert that a phenomenon admits a classical description (is classical-like) or admits a quantum description (is quantum-like), or neither. If that is true, then the idea of a classical-quantum boundary is a misnomer. There is no boundary, no scale, on one side of which all phenomena are quantum and the other side of which all phenomena are classical. We have seen that the assumption that being macroscopic is an indicator of classicality is simply false. There are macroscopic entities which possess many characteristics similar to those of quantum mechanical entities, including contextuality, organisms being the quintessential examples. There are macroscopic phenomena, such as psychological states, which manifest contextuality, discreteness, and even superpositions, and may even be describable using the mathematics of operators on Hilbert spaces, the language of quantum mechanics, if not the exact theory. Neural and collective intelligence systems may also exhibit quantum-like phenomena, though that is still a conjecture at this point. There are also macroscopic phenomena such as information flow in genetics, which is inherently describable by neither classical nor quantum mechanics as it lacks the most basic property, a Hamiltonian, from which a dynamic can be derived. The macroscopic world is simply too dynamically diverse for such a prosaic notion as a classical-quantum boundary. Moreover, there are abundant examples of inanimate complex systems which exhibit emergent phenomena and thus escape such a simplistic dichotomy [14,15,16].

It is possible to frame the question of a classical-quantum boundary more precisely without a simple reduction to a question of scale: What are the necessary and sufficient features that a system must possess in order for it to be described within either a classical or quantum framework? Framed in this manner, attention should shift from a fixation with scale, and instead focus on the intrinsic characteristics and dynamics of the system under consideration, and of its properties, which make it describable by a particular theory. To facilitate such an analysis, it would be helpful to have a general language within which such questions may be framed and discussed. This language might play the role of the cellular automaton frame in the Game of Life, upon which other frames supervene. It might play a role analogous to the universal dynamical system, a language of dynamics within which other dynamical languages may be framed. The study of semantic frames led to an investigation of different abstract systems for representing frames, including the idea of the reality game [17,44] and the causal tapestry [54]. Further study led to the creation of the Process Algebra as a candidate language. There are other such languages under development, one of the more prominent being the Functional Constructivism (FC) approach of Trofimova [20,21]. The focus here is on the Process Algebra because it more readily maps to more traditional formal languages (vector spaces, Hilbert space, manifolds) but the FC approach does offer much to say in terms of universal principles of behavioral and state construction (somewhat akin to the basic principles of computation used to formulate computation theory). The Process Algebra was developed as a response to Rosen, who suggested that the principles underlying biological systems were much more general than those derived from the study of inanimate matter [19].

## 5. The Play’s the Thing

The Process Algebra is a generative theory of reality, reflecting the characteristics of organisms as described previously [20,21,22,23,24,25,26,27,28]. Based on Whitehead [55,56], it takes the concept of becoming and places it first and foremost. Being is considered as secondary to becoming. The physical entities that we are and observe are emergent from a lower level of information-laden primitives called actual occasions (or informons in the Process Algebra). Just as neural and collective intelligence systems generate their behaviors anew each time [6,18,20,28], so does reality generate its entities. These actual occasions come into being under the action of process, they exist long enough to pass their information on to the next generation of actual occasions, and then fade from existence. The collection of actual occasions generated by a process in a single generation cycle (called a causal tapestry) is finite in number and possesses a causal ordering together with a causal metric. This makes it compatible with special relativity. Successive generations of causal tapestries are causally ordered and form the history of the system (although only the prior and nascent generations together with their generating process transiently exist, forming a compound present). The flow of information from the prior to nascent generation is causal and defines the causal structure linking tapestries. Note also that actual occasions do not move—they merely come into and fade out of existence. Information “flows” but not literally—more accurately, it is incorporated and dissipates.

The interested reader should look to the Appendix A for details about the Process Algebra and model. Briefly, a process generates informons, which form a causal tapestry. Given a causal tapestry, C, each of its informons, *n*, possess information, Γ_n_. These informons are then interpreted by some observer (akin to establishing a frame of reference) by embedding them into a causal manifold *M* ($n\to m_{n}$) and forming an emergent wave function via interpolation: $\phi_{C}(z)=\Sigma_{n}\Gamma_{n}\phi_{n}(z)$. The information “flow” takes place under a propagator *K* so that $\Gamma_{n}=\Sigma_{m}\Gamma_{m}$ where the sum is over the prior causal tapestry, similar to a Feynman propagator.

The generative nature of processes makes their actions contingent and context-dependent (due to interactions with other processes) and thus the appearance of probabilistically incompatible contexts (hence contextuality [22]) becomes commonplace.

In previous publications ([46,57], see Appendix A), it has been suggested that entanglement forces the processes which generate the individual participants to interact so as to create a single process which generates all participants concurrently. This single process generates correlations which violate the Bell inequalities in the absence of non-local influences by acting as a common cause [58,59]. This is consistent with other approaches which have suggested that the Bell inequalities are not about non-locality but rather about contextuality [60,61,62].

There are several problems with the Hilbert space formulation of quantum mechanics. The first, described previously, is that the use of an unqualified sum to form solutions to dynamical equations risks mixing distinct ontological entities, leading to much ontological confusion. The second is that physical entities are attributed states which extend throughout all of space. Together with a lack of constraints on the upper (and sometimes lower) limits on observables, this leads to the presence of divergences. The third is that little attention is paid to the fine details of initial conditions. That statement will likely raise howls of protest, since the usual boundary conditions are time 0 and some distant spatial boundary (usually infinity). However, these boundary conditions merely select out the set of eigenfunctions for the solution space and assuming separability of space and time, a general solution takes the form $\Sigma_{E}e^{\mp iEt}A_{E}\Psi_{E}\left(x\right)$ for a discrete set of E and constants A_E_. For the most part, the choice of constants is arbitrary once the set of eigenvalues has been determined. Moreover, note that in a time-independent setting, the Ψ_E_ are fixed for all times. The spatial structure simply *is,* extending across all of space and for all time.

The mathematical language of the Hilbert space simply lacks nuance; the Process Algebra, on the other hand, is able to maintain ontological distinctions, is finitary, and so avoids divergences, is local, and hence relativistically invariant, and generative, and so forces attention onto the processes by which physical entities come into and fade out of existence. It also forces attention onto how information flows in physical systems, not merely its presence in some abstract sense. The Hilbert space structure can be obtained as an asymptotic limit as the number of informons grows to infinity and the spatiotemporal scale shrinks to 0, the continuum limit (see Appendix A), and thus provides not the final theory, but merely an effective theory.

The discreteness of fundamental events appears forced upon us by two broad considerations. First of all, the use of continuous models, especially for potentials, invariably involves functions with an inverse dependence on distance. As a result, divergences are inevitable. The use of arbitrary cutoffs or elaborate renormalization schemes, while they do sweep the problems under the rug, seems unsatisfactory in the long run. As well, theories such as general relativity appear to be non-renormalizable, so the fix does not work. Second, arguments from the study of quantum gravity seem to suggest that at the smallest scales of space (Planck length) and time (Planck time), the continuous nature of space-time breaks down [39].

If space-time is discrete, then the next question is whether the actual occasions associated with a physical entity are finite or arbitrarily large (infinite). The wave function for a spherical wave diminishes by at least the order of 1/r as the radius r increases [60]. Following the principle that something is said to exist if and only if it makes a difference [48], and given that at large distances the size of the wave function is negligible, and thus makes no observable difference, it is therefore reasonable to conclude that the wave function simply does not exist at large distances. Models for a hydrogen atom confined within a spherical box [63,64] support this view. These studies show that the radial part of the wave declines rapidly as r increases, as does the correction to the free electron wave function which enclosing the atom induces. In fact, by 5.8 a.u. (atomic units, length) the difference is negligible and diminishes rapidly with increasing radius. This suggests that by 5.8 a.u. the wave effectively does not exist since enclosing it in a box at that distance makes little to no difference to the wave function.

## 6. You Cannot Get There from Here

If space-time is discrete, and the number of generated informons is finite, then the volume of space occupied by the informons in a causal tapestry must also be finite and must therefore have a finite maximal diameter, call this *d*(*P*). In the Process Algebra, two independent processes *P*_1_ and *P*_2_ are represented as the superordinate process *P*_1_, *P*_2_. In order for these two independent processes to be truly independent, the informons which they generate must be spatially separated from one another. If the distance between the informons of *P_1_* and *P*_2_ is less than the diameter of either process, then potentially during the act of generation, either process could attempt to generate an informon at the same spatiotemporal location as the other, resulting in an ontological conflict. This must be avoided. Thus, at the spatial regions within which these processes are generating informons, there comes a point where their spatial extents overlap, at which point the superordinate process must transition from *P*_1_, *P*_2_ to *P*_1_ ⊗ *P*_2_. This may seem like a trivial difference, but in the first case there is no interaction between the two processes while in the second case there is, since their generative actions have now become weakly coupled. Such a coupling does not affect the properties of either process, only their generative activities. If an interaction occurs in which some of the properties of either process are altered, then this is denoted by *P*_1_ ⊠ *P*_2_ to indicate that a strong coupling has been initiated.

Information can propagate to nascent informons only if there is a free path along which this may take place. This is akin to percolation of a substance within some other structure where it can only flow to unoccupied sites. If there is no free path, there is no information flow. Information also flows locally and causally.

Conversely, suppose one begins with a single process *P*, for example a free particle. Assume that over time some of the informons being generated by *P* separate spatially from the rest by a distance which is greater than can be covered by light in a generation cycle time (usually considered as Planck time). In such an event, these two groups of informons can no longer exchange information, although if the informons within each group lie close together, they can exchange information among themselves. In that case the original process naturally splits into two subprocesses *P*_1_, *P*_2_, one for each group. In this case the original process now splits, and this is represented by *P_1_*
$\hat{\hat{⊕}}$
*P*_2_. The free sum is used here because the subdivision is within a single process. This happens, for example, when light passes into the two arms of an interferometer. Here the spilt is not an arbitrary choice of the observer but rather arises intrinsically as a result of the internal dynamics of the generation process.

In the case that a system lies in a superposition of states, each individual state will be represented by its own process *P*_1_, *P*_2_, and the superordinate process will be denoted as *P* = *P_1_* ⊕ *P*_2_. Here, the exclusive sum must be used because these two subprocesses, while being involved in the generation of the same entity, are nevertheless ontologically distinct, generating distinct and disjoint collections of informons which spatiotemporally interleave, as do their interpolated wave functions. The interpolated wave function for the superordinate process is just the sum of the wave functions for the subprocesses. Thus, at the level of the actual occasions, these subprocesses are ontologically distinct, but at the level of the wave function this information is lost—it is this information which is missing from the Hilbert space formulation. Information from one subprocess may not enter into the generation of informons from the other process, but the generation of informons is sequential between the two subprocesses and shifts from one to the other can only occur if there is a free path from the informons of one to those of the other. If no such path exists then it is not possible to shift to the other subprocess, and it simply ceases. This is the mechanism of so-called wave function collapse—interaction of one subprocess with a measurement process effectively blocks paths to the remaining subprocesses, and so they cease to be active. Any process becomes inactive when there are no longer free paths along which information may flow.

The existence of ontologically distinct superposition states reduces to the question of whether or not there are persistent free paths which enable the superordinate process to effect a flow from one subprocess to another. Measurement may obstruct this flow. However, there is, a priori, no reason why internal dynamics could not result in an obstruction to information flow, which would either spontaneously terminate any superposition or prevent its formation in the first place. Again, these subtleties are lacking in the Hilbert space formulation.

From the Process Algebra perspective then, what is required if a superposition is to take place? It means that there is a larger process generating a large number of informons collectively transitioning between two macrostates. Those macrostates must be able to be defined in terms of configurations of microstates. And *all* of the subprocesses must be correlated.

Suppose we consider just a pair of processes and write the superposition as
$$\left(P_{1}⊗P_{2}\right)⊕\left(P_{1^{\prime }}⊗P_{2^{\prime }}\right)$$

This implies that the processes must toggle back and forth between (P_1_
⊗ P_2_) and (P_1′_
⊗ P_2′_).

Moreover, each component subprocess must be free to toggle back and forth between their states so we must have (P_1_) ⊕ (P_1′_) and (P_2_) ⊕ (P_2′_), if they could be viewed separately. Note that if we allowed the two systems to be joined from these separate superpositions then we would obtain [(P_1_) ⊕ (P_1′_)] ⊗ [(P_2_) ⊕ (P_2′_)] = [(P_1_) ⊗ (P_2_)] ⊕ [(P_1′_) ⊗ (P_2_)] ⊕ [(P_1_) ⊗ (P_2′_)] ⊕ [(P_1′_) ⊗ (P_2′_)]. Clearly, this means that the original processes were entangled in order to create the original superposition. In order to maintain the superposition it is then necessary to maintain the entanglement.

As the number of processes increases, it becomes unwieldly to try to represent them with a long list of pairwise interactions, especially when attempting to define superordinate processes. Instead, these interactions may be represented by means of (directed) graphs, with vertices labelled by processes and edges by particular interaction subtypes. Forming a superposition between two process states means forming a superposition between two of these graphs. Teasing out the various possibilities is a work in progress, but the discussion above illustrates that there will be a myriad of entanglements across the set of processes, and these must be sustained across the entire set of processes, and moreover, if these states are dynamic, they must also be sustained throughout the sequence of transitions that form the dynamical evolution of the system. It is quite possible within such a tangled graph of connections that an entanglement within one set of processes might run afoul of entanglements within other sets, especially whenever these two sets overlap over some processes.

While the details need to be worked out carefully, the above discussion suggests two points. First, that superpositions will be possible only when there are free paths which can enable the toggling of informons between the processes generating the components of the superposition, and second, that entanglements between the processes forming each component of the superposition must be consistent for the entire set of subprocesses involved. This is a strong condition. Size alone need not be the important criterion here. The ability to form a superposition will crucially depend upon how the subprocesses of the two superposition component processes are configured.

## 7. No Cats Allowed

Finally, consider the situation of Schrodinger’s cat. The cat is placed, alive, in an isolated room together with the usual radioactivity-triggered death device. The presumption is that the cat becomes entangled with the killing machine, and now exists in a superposition state of alive and dead states. It is never stated or explained how or why this superposition occurs, apart from it being a reflection of a lack of knowledge on the part of the observer. Framed solely in terms of the observer’s knowledge, there is no problem. That knowledge will be indeterminate until such a time as the observer opens the room and checks to see whether the cat is alive or not. At that point their knowledge becomes definite. However, there is nothing in that scenario to make any statement about the state of the cat prior to the observation. The lack of knowledge on the part of the observer need have no bearing at all upon the relationship between the cat and the death trap, which is the only thing that matters to the cat, if it is aware of its peril at all.

Consider what is involved here. A living cat is an extraordinary complex of interacting processes extending from the level of fundamental particles all the way to the level of the cat as a whole. The likely presence of emergence means that we cannot take for granted that superordinate processes can be simply formed from interactions among lower processes. Like in the Game of Life, some higher level process may exist only when special configurations of lower level processes are in place. Endless random and nonrandom factors could disrupt those configurations, resulting in the cessation of the superordinate process. Moreover, the physiological processes that make a cat must unfold in complex coordination with other processes, and the temporal coherence of this coordination must be maintained through the life of the cat. The graph of a living cat is expected to be strongly connected, with levels and levels of superordinate processes and interactions, with complex temporal linkages. Even the components of which a cat is comprised are in flux, even when the cat is simply sitting still. It must respire and perspire and generate urine and food waste. It metabolizes and creates heat.

A dead cat, on the hand, lacks almost all of those complex processes. A dead cat is essentially a conglomeration of inert components, which are no longer interacting apart from simple local thermal effects. A dead cat does not engage in even the simplest of metabolic processes. There is no flux of components (at least for a few hours). Although the description of a dead cat in Process Algebra terms cannot be spelled out, it should be clear that the graphical structure of subprocess interactions will primarily involve simple nearest neighbor interactions and an absence of superordinate processes and interactions.

The bottom line is that the process corresponding to a live cat bears no relationship to that of a dead cat. There is simply no way in which one can form a superposition of these two states since they are not actually two states of one process, they are two completely different processes. There is a path from the process that is a live cat to the process that is a dead cat. However, there is no path taking the cat from dead to alive. The absence of any such path means that a superposition of these two processes cannot take place. Merely placing a cat in a room does not create a superposition. This is another deep failing of the Hilbert space approach because it deals only with epistemology and utterly leaves out the essential ontology of situations.

## 8. Conclusions

An examination of the classical-quantum dichotomy through the lens of the Process Algebra framework suggests that it is really a false dichotomy. Many of the conundrums associated with the classical and quantum worlds really arise because of the particular choice of mathematics used to describe them: the language of finite dimensional vector spaces and function on the one hand, and Hilbert spaces and operators on the other hand The failure of both approaches to take into account issues of initial conditions and the generation of physical entities leaves these systems incomplete, and thus prone to misrepresentation and misinterpretation. The Process Algebra is one step towards a unifying language and a complete theory. It shows that the boundary between classical and quantum is not hard, rather it is an expression of subtle differences in the way in which different physical entities are formed, and in their dynamics. We can speak of classical-like objects, classical-like organisms, quantum-like entities, but not of two disparate domains. Reality is unitary, but nuanced, and the Process Algebra helps to better capture these nuances.

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
