# Peer review of "The Classical-Quantum Dichotomy from the Perspective of the Process Algebra"

_entropy, 2022, doi:10.3390/e24020184_

Round 1
Reviewer 1 Report
This paper is a very interesting piece of work that presents the classical-quantum boundary (dichotomy in author’s wording) from a promising perspective that might result very fruitful in gaining insight on the long-lasting quantum conundrums and producing a meaningful progress in the foundations of quantum mechanics. Therefore I recommend the publication of this article on which I proceed to enumerate some minor remarks:
1) In the line 116, it lacks the end of the paragraph.
2) In the line 125, a symbol in the equation is not defined.
3) Finally, I suggest to briefly discuss the inherent difficulties in the definition of complexity (when complex systems are first mentioned in the main text, in line 155).
Author Response
Reviewer 1
This paper is a very interesting piece of work that presents the classical-quantum boundary (dichotomy in author’s wording) from a promising perspective that might result very fruitful in gaining insight on the long-lasting quantum conundrums and producing a meaningful progress in the foundations of quantum mechanics. Therefore I recommend the publication of this article on which I proceed to enumerate some minor remarks:
1) In the line 116, it lacks the end of the paragraph.
2) In the line 125, a symbol in the equation is not defined.
Thank you for pointing out these errors. They have been corrected.
3) Finally, I suggest to briefly discuss the inherent difficulties in the definition of complexity (when complex systems are first mentioned in the main text, in line 155).
A brief section has been added pointing out some of the difficulties with this definition and pointers to my favorite literature on the matter.
Reviewer 2 Report
The manuscript "The Classical-Quantum Dichotomy from the Perspective of the Process Algebra" (ID: entropy-1561711) critiques standard attempts to define the distinction between classical and quantum physics, and purports to provide insight into this critique employing considerations from the process algebra formalism.
The manuscripts begins strongly in the first 2 sections setting out standard attempts to define the classical-quantum dichotomy and diagnosing why any such sharp distinction will ultimately fail. This analysis is interesting and likely to generate further discussion in the literature. Section 3 introduces the concept of 'semantic frames' to illuminate the problems outlined in the first 2 sections and, again, this is a highly interesting, intuitive, and illuminating analysis.
But the logical flow of the manuscript begins to break down towards the end of this section, and the manuscript suffers as a result. I think that there needs to be a stronger and more explicit connection between the discussion of classical-quantum dichotomy and the subsequent material from Section 4 onwards on the process algebra. As it stands, it seems like a manuscript consisting of two independent discussions. As an example, so much is made of the difference between ontological and epistemological readings of the classical-quantum dichotomy in the opening few pages, but then these two approaches are mentioned again only briefly at the end of Section 6. What's the importance of this setup of the problem? More needs to be made explicitly regarding how the 'perspective of the process algebra' impinges upon the critique of the classical-quantum dichotomy (which appears to be the fundamental purpose of the manuscript).
While I am not an expert in the process algebra, so cannot comment on the technical aspects/correctness of Sections 4 and 5, it seems that the example of the Game of Life in Section 3 is supposed to be acting as a key analogy or, at the very least, intuition pump for how we should understand what the process algebra suggests could be the dynamical structure of reality. I think this would be a very interesting proposal, and worthy of publication. But more needs to be said explicitly about this. As a result, I recommend this manuscript be accepted for publication, on the proviso that the author can suitably develop a stronger, more explicit connection between the earlier analysis of the classical-quantum dichotomy and the process algebra.
A couple of minor comments:
(i) The terms 'ontological' and 'epistemological' are defined in the opening paragraphs and applied to the distinction between classical and quantum. There is an element of 'epistemological' that is missing from this definition. Not only does epistemological refer to the formal languages and theories we use to describe entities, but it also refers to the way we discover and find out about entities, and this is an important part of the (supposed) distinction between classical and quantum. This essentially forms part of the ensuing discussion, when the author discusses 'measurement', but it might be good to be explicit about this in the definition of 'epistemology'.
(ii) The first paragraph at the top of page 10 is a bit quick. In particular, it's not clear what the following sentence is supposed to imply, exactly: "The appearance of Bell correlations in the absence of non-local influences occurs through interactions among processes [48], which serve as common causes [49]." This is a bold claim and requires more than the cursory justification it is currently given.
And some typos:
- p.3, l.116: The end of the paragraph ends mid-sentence "Another feature of quantum entities is that measurements..."
- p.4, ~l.164: ref [16] is missing from somewhere in here
- p.4, l.178: "insert" should be "inert"
- p.5 l.214: "presence of absence" should be "presence or absence"
- p.6, l.264: "sources if distinction" should be "sources of distinction"
- p.7, l.315: "which as themselves" should be "which are themselves"
- p.7, l.317: is ref [35] missing from this sentence?
- p.7, l.333: "Wee" should be "we"
- p.7, l.339: "there exist having" might be "there exist some having"?
- p.8, l.360: "mechanics are capable" should be "mechanics is capable"
- p.8, l.363: delete extra "systems." after ref
- p.8, l.377: "mathematics by rather" should be "mathematics but rather"
- p.8, l.382: "but in each" should be "but each"
- p.9, l.438: "be analogous that between" should be "be analogous between"
- p.9, l.443: "organisms are described" should be "organisms as described"
- p.9, l.451: "by process" should be "by a process"
- p.10, l.468: "with other process" should be "with other processes"
- p.10, l.517-18: is "a.u." supposed to be 'astronomical units'?
- p.11, l.531: "This as the" might be "Thus at the"?
- p.11, l.533: paragraph seems to end mid-sentence again (or maybe needs the next line to join back up to it)
- p.12, l.578: "spontaneously terminated" should be "spontaneously terminate"
- p.12, l.591: "if they could viewed" should be "if they could be viewed"
- p.12, l.592: paragraph ends mid-sentence again
- p.12, l.599: "with long list" should be "with a long list"
- p.12, l.601: "with vertices been labelled" should be "with vertices labelled"
- p.12, l.609: "these two set overlap" should be "these two sets overlap"
- p.12, l.612: "free paths enables" could be "free paths enabled"?
- p.13, l.624: "apart it" should be "apart from it"
- p.13, l.626: "will indeterminate" should be "will be indeterminate"
- p.13, l.641: "life of cat" should be "life of the cat"
- p.13, l.651: "interactions into" might have a verb missing ("interactions [verb missing] into")
- p.14, l.697: "actions of process" should be "actions of a process"
- p.15, l.759: extra full stop
- p.15, l.772: "that the neither" should be "that neither"
Author Response
Reviewer 2
The manuscript "The Classical-Quantum Dichotomy from the Perspective of the Process Algebra" (ID: entropy-1561711) critiques standard attempts to define the distinction between classical and quantum physics, and purports to provide insight into this critique employing considerations from the process algebra formalism.
The manuscripts begins strongly in the first 2 sections setting out standard attempts to define the classical-quantum dichotomy and diagnosing why any such sharp distinction will ultimately fail. This analysis is interesting and likely to generate further discussion in the literature. Section 3 introduces the concept of 'semantic frames' to illuminate the problems outlined in the first 2 sections and, again, this is a highly interesting, intuitive, and illuminating analysis.
But the logical flow of the manuscript begins to break down towards the end of this section, and the manuscript suffers as a result. I think that there needs to be a stronger and more explicit connection between the discussion of classical-quantum dichotomy and the subsequent material from Section 4 onwards on the process algebra. As it stands, it seems like a manuscript consisting of two independent discussions. As an example, so much is made of the difference between ontological and epistemological readings of the classical-quantum dichotomy in the opening few pages, but then these two approaches are mentioned again only briefly at the end of Section 6. What's the importance of this setup of the problem? More needs to be made explicitly regarding how the 'perspective of the process algebra' impinges upon the critique of the classical-quantum dichotomy (which appears to be the fundamental purpose of the manuscript).
That you for your comments. Rereading the manuscript I can see why you would feel this to be the case. I did not properly motivate the rationale for introducing the Process Algebra framework. I have expanded the discussions in the preceding sections and I have added a new section to serve as a bridge linking the two halves. I hope that this makes the relationship between the two halves clearer to the reader.
While I am not an expert in the process algebra, so cannot comment on the technical aspects/correctness of Sections 4 and 5, it seems that the example of the Game of Life in Section 3 is supposed to be acting as a key analogy or, at the very least, intuition pump for how we should understand what the process algebra suggests could be the dynamical structure of reality. I think this would be a very interesting proposal, and worthy of publication. But more needs to be said explicitly about this. As a result, I recommend this manuscript be accepted for publication, on the proviso that the author can suitably develop a stronger, more explicit connection between the earlier analysis of the classical-quantum dichotomy and the process algebra.
I have expanded this section as well to better link the Life discussion to the semantic frame concept and the analogy to QM and CM.
A couple of minor comments:
- The terms 'ontological' and 'epistemological' are defined in the opening paragraphs and applied to the distinction between classical and quantum. There is an element of 'epistemological' that is missing from this definition. Not only does epistemological refer to the formal languages and theories we use to describe entities, but it also refers to the way we discover and find out about entities, and this is an important part of the (supposed) distinction between classical and quantum. This essentially forms part of the ensuing discussion, when the author discusses 'measurement', but it might be good to be explicit about this in the definition of 'epistemology'.
Text was added to include these additional issues under the rubric of epistemology and to clarify their place in the ensuing discussion.
- The first paragraph at the top of page 10 is a bit quick. In particular, it's not clear what the following sentence is supposed to imply, exactly: "The appearance of Bell correlations in the absence of non-local influences occurs through interactions among processes [48], which serve as common causes [49]." This is a bold claim and requires more than the cursory justification it is currently given.
I am aware that it is bold but I have provided some references in the literature where I developed the claim further – I did not consider this paper to be the place to repeat all of those arguments.
And some typos:
- p.3, l.116: The end of the paragraph ends mid-sentence "Another feature of quantum entities is that measurements..."
- p.4, ~l.164: ref [16] is missing from somewhere in here
- p.4, l.178: "insert" should be "inert"
- p.5 l.214: "presence of absence" should be "presence or absence"
- p.6, l.264: "sources if distinction" should be "sources of distinction"
- p.7, l.315: "which as themselves" should be "which are themselves"
- p.7, l.317: is ref [35] missing from this sentence?
- p.7, l.333: "Wee" should be "we"
- p.7, l.339: "there exist having" might be "there exist some having"?
- p.8, l.360: "mechanics are capable" should be "mechanics is capable"
- p.8, l.363: delete extra "systems." after ref
- p.8, l.377: "mathematics by rather" should be "mathematics but rather"
- p.8, l.382: "but in each" should be "but each"
- p.9, l.438: "be analogous that between" should be "be analogous between"
- p.9, l.443: "organisms are described" should be "organisms as described"
- p.9, l.451: "by process" should be "by a process"
- p.10, l.468: "with other process" should be "with other processes"
- p.10, l.517-18: is "a.u." supposed to be 'astronomical units'? It refers to atomic units of length
- p.11, l.531: "This as the" might be "Thus at the"?
- p.11, l.533: paragraph seems to end mid-sentence again (or maybe needs the next line to join back up to it)
- p.12, l.578: "spontaneously terminated" should be "spontaneously terminate"
- p.12, l.591: "if they could viewed" should be "if they could be viewed"
- p.12, l.592: paragraph ends mid-sentence again
- p.12, l.599: "with long list" should be "with a long list"
- p.12, l.601: "with vertices been labelled" should be "with vertices labelled"
- p.12, l.609: "these two set overlap" should be "these two sets overlap"
- p.12, l.612: "free paths enables" could be "free paths enabled"?
- p.13, l.624: "apart it" should be "apart from it"
- p.13, l.626: "will indeterminate" should be "will be indeterminate"
- p.13, l.641: "life of cat" should be "life of the cat"
- p.13, l.651: "interactions into" might have a verb missing ("interactions [verb missing] into")
- p.14, l.697: "actions of process" should be "actions of a process"
- p.15, l.759: extra full stop
- p.15, l.772: "that the neither" should be "that neither"
Thank you for your careful attention to my manuscript. This must have been quite time consuming, and I am grateful for you saved me a lot of work. I believe that I have corrected all of the problems that you have noted. Hopefully the manuscript reads much better now.